# Experiences and concerns of health workers throughout the first year of the COVID-19 pandemic in the UK: A longitudinal qualitative interview study

Aleksandra J. Borek[1‡]*, Caitlin Pilbeam[1‡], Hayley Mableson[2], Marta Wanat[1], Paul Atkinson[3], Sally Sheard[3], Anne-Marie Martindale[2], Tom Solomon[2,4,5], Christopher C. Butler[1], Nina Gobat[1], Sarah Tonkin-Crine[1,6]

1 Nuffield Department of Primary Care Health Sciences, University of Oxford, Oxford, United Kingdom, 2 Institute of Infection, Veterinary and Ecological Sciences, University of Liverpool, Liverpool, United Kingdom, 3 Institute of Population Health, University of Liverpool, Liverpool, United Kingdom, 4 National Institute for Health Research Health Protection Research Unit in Emerging and Zoonotic Infections, University of Liverpool, Liverpool, United Kingdom, 5 Department of Neurology, Walton Centre NHS Foundation Trust, Liverpool, United Kingdom, 6 National Institute for Health Research Health Protection Research Unit in Healthcare Associated Infections and Antimicrobial Resistance, Oxford, United Kingdom

‡ AJB and CP authors contributed equally to this work are joint first authors.
* Aleksandra.borek@phc.ox.ac.uk

**Data Availability Statement:** Data excerpts are provided within the paper and further can data be provided from the authors on reasonable request.

## Abstract

### Objective

To identify the experiences and concerns of health workers (HWs), and how they changed, throughout the first year of the COVID-19 pandemic in the UK.

### Methods

Longitudinal, qualitative study with HWs involved in patient management or delivery of care related to COVID-19 in general practice, emergency departments and hospitals. Participants were identified through snowballing. Semi-structured telephone or video interviews were conducted between February 2020 and February 2021, audio-recorded, summarised, and transcribed. Data were analysed longitudinally using framework and thematic analysis.

### Results

We conducted 105 interviews with 14 participants and identified three phases corresponding with shifts in HWs' experiences and concerns. (1) Emergency and mobilisation phase (late winter-spring 2020), with significant rapid shifts in responsibilities, required skills, and training, and challenges in patient care. (2) Consolidation and preparation phase (summer-autumn 2020), involving gradual return to usual care and responsibilities, sense of professional development and improvement in care, and focus on learning and preparing for future. (3) Exhaustion and survival phase (autumn 2020-winter 2021), entailing return of changes in responsibilities, focus on balancing COVID-19 and non-COVID care (until becoming overwhelmed with COVID-19 cases), and concerns about longer-term impacts of

The full dataset is not publicly available to protect participants' anonymity. Queries regarding data access may be directed to the authors or to datasecurity@phc.ox.ac.uk.

**Funding:** Funding was provided by the UKRI/NIHR 2019 nCoV Rapid Response Call through a grant (Grant No. NIHR200907) awarded to SS, TS, NG and STC, and which also supported AJB, CP, HM, PA, A-MM, with support from the National Institute for Health Research (NIHR) Health Protection Research Unit in Emerging and Zoonotic Infections at University of Liverpool in partnership with Public Health England, and in collaboration with Liverpool School of Tropical Medicine and the University of Oxford. MW is supported by the EU Horizon 2020 Research and Innovation programme (Grant No. 101003589). SS is supported by the Wellcome Senior Investigator Award and by the NIHR Health Protection Research Unit in Emerging and Zoonotic Infections (Grant No. NIHR200907). TS is supported by the NIHR Health Protection Research Unit in Emerging and Zoonotic Infections (Grant No. NIHR200907), NIHR Global Health Research Group on Brain Infections (No. 17/63/110), the UK Medical Research Council's Global Effort on COVID-19 Programme (MR/V033441/1), and COVID-19 Clinical Neuroscience Study (COVID-CNS, MR/V03605X/1). ST-C is supported by a grant from the NIHR Health Protection Research Unit in Healthcare Associated Infections and Antimicrobial Resistance (NIHR200915).

**Competing interests:** AJB, CP, HM, MW, PA, AMM, CCB, NG and STC declare no competing interests. SS is a member of the Liverpool Health Protection Board and the Silver Command for Liverpool COVID-19 management. TS was Chair/Co-Chair of the United Kingdom Research and Innovation / National Institute for Health Research COVID-19 Rapid Response and Rolling Funding Initiatives, was an Advisor to the UK COVID-19 Therapeutics Advisory Panel and is a member of the UK Medicines and Healthcare Products Regulatory Agency COVID-19 Vaccines Benefit Risk Expert Working Group. TS holds shares in Medefer Limited.

**Abbreviations:** COVID-19, coronavirus disease caused by SARS-CoV-2; GP, general practitioner (family physician; HCW, healthcare worker; ICU, intensive care (treatment) unit; NHS, National Health Service in the UK; PPE, personal protective equipment; SARS-CoV-2, severe acute respiratory syndrome coronavirus 2.

unceasing pressure on health services. Participants' perceptions of COVID-19 risk and patient/public attitudes changed throughout the year, and tiredness and weariness turned into exhaustion.

## Conclusions

Results showed a long-term impact of the COVID-19 pandemic on UK HWs' experiences and concerns related to changes in their roles, provision of care, and personal wellbeing. Despite mobilisation in the emergency phase, and trying to learn from this, HWs' experiences seemed to be similar or worse in the second wave partly due to many COVID-19 cases. The findings highlight the importance of supporting HWs and strengthening system-level resilience (e.g., with resources, processes) to enable them to respond to current and future demands and emergencies.

## Introduction

The global spread of a novel coronavirus SARS-CoV-2 since late 2019, causing the disease COVID-19, has had a profound impact on societies, healthcare systems, and health workers (HWs) across the world. The first cases in the UK were confirmed on 31 January 2020 [1]. Since then, the daily numbers of people testing positive for, and dying from, COVID-19 increased in the first wave (March-May, peaking in April 2020) and in the second wave (September-January, peaking in November 2020 and January 2021) [2], leading to over 156,000 deaths in the UK by August 2021 [3]. HWs in the UK National Health Service (NHS) have been at the forefront of the response to the pandemic.

Previous research has highlighted the importance of qualitative research as a way to understand the experiences and evolving concerns of frontline HWs and patients, and help make improvements impacting on healthcare delivery [4–6]. Qualitative studies conducted with HWs in the UK during the early months of the pandemic identified concerns at the time; these included: inadequate access to personal protective equipment (PPE) and COVID-19 testing; rapidly changing and inconsistent guidelines; a lack of training; communication challenges; increased workload and rapidly changing work conditions [7–12]. Studies also highlighted the impact of pandemic-related stressors and work conditions on the mental health and wellbeing of HWs, both in the UK [12–14] and internationally [15–20]. Following the first wave of COVID-19, calls were made to learn lessons for future waves and pandemics [21–23]. Notwithstanding these calls in autumn 2020 the numbers of COVID-19 cases and deaths rose again, eventually exceeding those from the first wave [3], mounting further pressure on an overburdened healthcare system and HWs. HWs are a critical workforce and understanding their experiences and needs, and how to support them, is important not only from a moral perspective but also to enable ongoing functioning of health services.

The purpose of this study was to explore the experiences and concerns of HWs and how they evolved throughout the first year of the COVID-19 pandemic in the UK, with the intention to help recognise, reflect on, and use lessons from the COVID-19 pandemic for policy and practice revisions. This article makes a novel contribution by providing findings from a longitudinal, year-long study with the same group of HWs from different healthcare settings in the UK, and by illustrating *how* we may learn from the pandemic in this way over time.

## Methods

This was a longitudinal qualitative study using serial interviews; a study design that is particularly useful for investigating change over time and collecting more in-depth qualitative data [24]. The study was approved by the University of Oxford's Research Ethics Committee (ref. R69302).

### Participants

Participants were recruited using a snowballing approach through researchers' and collaborators' contacts with practicing clinicians. They were eligible if they were HWs involved in patient management or supporting delivery of care related to, or affected by, COVID-19. They were purposefully selected to ensure participants were from different primary and secondary care settings, professional roles, and seniority levels. Participation was voluntary, without monetary compensation. Participants were provided with study information, informed that they would be interviewed multiple times over a year (depending on their availability), and gave informed consent verbally for longitudinal involvement; a written record of each participant's consent was also made and retained. Participants were reassured that they could withdraw from the study at any time.

### Interviews

In-depth, semi-structured interviews were conducted between February 2020 and February 2021 by CP and HM (researchers experienced in qualitative research, with no prior relationship with participants) via telephone or video, as preferred by the participants. Interviews were repeated with the same participants throughout the year, depending on participant availability. A semi-structured topic guide (S1 Table) was used flexibly with questions exploring participants' changing experiences of clinical service adaptations and readiness, and perceptions of resilience and response. The interviews were flexible in terms of length and topics (emerging issues and concerns), allowing participants to raise and discuss issues perceived as important and relevant. They were audio-recorded and all were summarised by the researchers conducting the interviews. Based on the summaries and perceived data richness, we selected interviews for transcription and most were transcribed verbatim.

### Analysis

Data were analysed using thematic and framework methods, taking an essentialist/realist epistemological stance [25, 26]. Two approaches were used to analyse data longitudinally: trajectory (i.e. focused on change over time) and recurrent cross-sectional (i.e. focused on themes and changes at different time-points) [27]. The analysis was led by AJB who familiarised herself with the data by reading interview notes and transcripts. Emerging topics and potential themes were discussed with the research team periodically. After completing all interviews, CP wrote case summaries for each participant, summarising the main experiences, perceptions and concerns, and how they changed over time; this helped identify trajectories of individual experiences. Based on the summaries, key topics relevant to the research question were identified and agreed by the team. To identify similarities and differences across time and dataset, AJB created two frameworks summarising the data related to the key topics and time-points: per interview/participant and per setting. Transcripts and interview notes were uploaded to NVivo software (v.12, QSR International) and AJB coded them using the key topics as categories and more detailed codes to capture the details.

## Results

Interviews (n = 105, 4–10 per participant; 46 in February-May 2020, 30 in June-September 2020, 29 in October 2020-February 2021) were conducted with 14 HWs from general practice (3), hospital's accident and emergency departments (2) and hospitals (9) from different areas in the UK (Table 1). Interviews lasted between 20 and 80 (median 40) minutes (except one 8-minute interview when the participant was called to an emergency). Participant-approved pseudonyms and role descriptors are used for anonymity.

We identified three overlapping phases of the pandemic that corresponded with shifts in HWs' experiences and concerns: (1) emergency and mobilisation (late winter—spring 2020), (2) consolidation and preparation (summer—early autumn 2020), and (3) exhaustion and survival (late autumn 2020—winter 2021). We focus on the changing impact of the pandemic on HWs' experiences and concerns related to their professional roles and care provision, and personal health and wellbeing. Table 2 provides a summary of the sub-themes.

### Phase 1: Emergency and mobilisation

**Context.** This phase involved a rapid response by HWs and the healthcare system to the emerging pandemic. It meant rapidly changing ways of working and caring for patients, mostly by reducing contact to prevent transmission and freeing up resources for COVID-19 care (e.g., limiting/stopping non-essential care). It also involved dealing with insufficient pandemic preparedness (e.g., limited PPE), mobilising resources (e.g., redeployment), and significant uncertainties related to the unknown pathogen and unfamiliar situations. This contributed to increased stress from the perceived risks, changing roles, and multiple impacts on the health of patients, HWs and their families.

**Table 1. Participant characteristics.**

| Setting | Participants' pseudonyms[a] | Participants' roles[a] | Experience in the role[b] | Number of interviews | Geographical area |
|---|---|---|---|---|---|
| General practice | Nathan | GP, Clinical Director in Primary Care Network | Senior | 9 | Merseyside |
| | Arthur | GP Partner | Senior | 9 | Merseyside |
| | Fiona | GP | Early-career | 7 | North of Scotland |
| Emergency | Linda | Clinical Lead & Consultant, Emergency Department | Senior | 7 | Cheshire |
| | Kyle | Registrar, Accident & Emergency Department | Mid-career | 10 | Essex |
| Hospital | Nick | Clinical Director, Hospital Department | Senior | 9 | Merseyside |
| | Carmen | Specialist Nurse, redeployed to ICU | Senior | 8 | London |
| | Dylan | Specialist Nurse, redeployed to ICU | Mid-career | 7 | London |
| | Theresa | Specialist Nurse, redeployed to ICU | Senior | 5 | Essex |
| | Ruth | Retired Chief Nurse, redeployed to a COVID-19 testing service | Senior | 4 | Worcestershire |
| | Bonnie | Speech & Language Therapist | Mid-career | 7 | Essex |
| | Claire | Registered Specialist Dietician | Mid-career | 6 | London |
| | Taren | Trainee, Obstetrics & Gynaecology, redeployed to Covid-19 wards | Early-career | 9 | London |
| | Maggie | Neonatal Registrar | Mid-career | 8 | Scotland, then London |

Abbreviations: GP–General Practitioner, ICU- Intensive Care Unit.

[a] Participants authorised the use of pseudonyms and the descriptions of their roles in the publication.

[b] Early-career: in training or <2 years; mid-career: 2–10 years; senior: >10 years of experience in the role.

Table 2. Summary of key topics and sub-themes.

| Phase | HWs' experiences and main concerns related to: | | |
|---|---|---|---|
| | **Context** | **Professional roles & care** | **Personal health & wellbeing** |
| **Phase 1** (winter—spring 2020, wave 1) *Emergency & mobilisation* | • Wave 1, peaking in April 2020<br>• Rapid adaptations<br>• Mobilising resources<br>• Uncertainty about the new pathogen and the situation | • Shifting responsibilities & redeployment<br>• New skills & training required<br>• Challenges in patient care in an emergency | • Perceived risk of COVID-19 & impact on others<br>• Tiredness & emotional weariness<br>• Experiencing patient/public attitudes |
| **Phase 2** (summer—autumn 2020) *Consolidation & preparation* | • Moving from emergency state to (new) normal<br>• Learning lessons<br>• Preparing/planning | • Gradual return to usual care & responsibilities<br>• Sense of professional development & improvement in services & care<br>• Learning & preparing for future | • Frustration with working conditions<br>• Partial recuperation<br>• Concerns about risk of complacency & about future |
| **Phase 3** (autumn 2020—winter 2021, wave 2) *Exhaustion & survival* | • Wave 2, peaking in November 2020 and January 2021<br>• Sense of better preparation<br>• Trying to continue non-COVID care<br>• Overwhelmed by the numbers of COVID-19 cases | • Shifting responsibilities & redeployment<br>• Balancing COVID-19 & non-COVID care<br>• Concerns about long-term impact of unceasing pressure | • Changed perceptions of risk of COVID-19<br>• Exhaustion & resignation<br>• Experiencing changed patient/public attitudes & behaviour |

*Everything changed so quickly.* [Taren, May '20]

*It was just at all cost to try and keep these people alive if you can and so, yeah, it felt very–it felt how you could imagine being in a war zone, just get on with it.* [Dylan, June '20]

**Impact on roles and care.** *Shifting responsibilities and redeployment.* The emergence of the pandemic led to changes in participants' responsibilities. Those in senior roles focused on planning and managing the response, and reorganising care delivery, and stopped any non-priority tasks (e.g., providing specialty training). There was also a shift in the type of clinical work: non-urgent and routine care was paused, replaced by COVID-related care. General practitioners (GPs) were identifying and contacting clinically vulnerable patients instructed to self-isolate, and 'correcting' the lists of those notified centrally by the NHS; dealing with increasing psychological and social (e.g., domestic, alcohol-related) issues during a national lockdown; focusing on palliative care, do-not-resuscitate orders and concerns about ethical issues; and working in COVID-19 clinics/hubs. In emergency departments, clinicians reported shifting to fully clinical work, dealing with administrative and managerial work in their own/extra time. In hospitals, non-urgent care stopped, many patients were discharged, and HWs focused on preparing and then dealing with increasing numbers of patients with COVID-19.

In the absence of usual care, some HWs described having less work and some volunteered, or were selected, for redeployment to support COVID-19 and intensive care. Some participants described working with more responsibility (e.g., covering for shortages of more senior HWs), and some described feeling like they moved into junior roles due to the lack of specific skills needed in the area in which they were redeployed.

*I've gone from being quite senior and confident in my role and able to do my job quite comfortably to feeling like a student nurse again. (. . .) I wonder if they could have provided some training to upskill me so I could have been more useful in the clinical area that I am redeployed now.* [Carmen, April '20]

Those redeployed to intensive care units (ICUs) or COVID-19 wards described significant pressure and stress related to inadequate communication and training, unpredictability of

work tasks, and unfamiliarity with the team and environment. For example, Carmen described how, after volunteering for redeployment in a Nightingale hospital and attending required training, she received no communication and ended up waiting at home, feeling anxious about loss of pay, until she asked to return to her previous ward.

*New skills and training required*. Changed ways of working, social distancing, infection prevention measures, and the new focus of participants' roles meant that they required new skills and training. GPs required new skills for managing patients remotely, describing concerns about making clinical decisions, risks of missing serious illnesses, and inability to refer patients to specialists. Participants from emergency departments and hospitals reported some (but insufficient) training related to COVID-19 and found managing patients with COVID-19 difficult due to multiple symptoms and little evidence on management. Those redeployed to ICUs perceived training for their new role as inadequate, that their existing skills were not utilised, and that they lacked specialist skills (e.g., using ventilators) which made them depend on guidance from busy ICU nurses which potentially posed risks to patients.

> *. . .we were given four hours [of training] and then two supernumerary days and then you were expected to take [a] patient. Now I take an intensive care patient on ventilation and [Extracorporeal membrane oxygenation] when I've never had experience in looking after those, so I think people were just worried about the fact that they could possibly harm others, just having to deal with the stress and pressure of essentially keeping someone alive when you don't really. . . it's sort of like flying a plane and not knowing what to do.* [Dylan, April '20]

Participants also described challenges of communicating with families about end of life care and deaths by telephone and in PPE, which was new to their role.

*Challenges in patient care*. With reduced contact from patients and suspension of non-urgent care, GPs were concerned about a build-up of unaddressed health issues and how this would be judged in the future. Those in emergency departments described being used to provide care under pressure but then felt helpless with little that they could do to help so many patients and staff with COVID-19 symptoms. They also reported keeping patients longer in the emergency department rather than admitting them to reduce the demand for beds.

In hospitals, nurses were concerned about not being able to provide as high a standard of care during the pandemic as they would like and finding it difficult to prioritise patient dignity; for example, due to caring for multiple patients simultaneously or time pressure.

> *. . .it definitely felt that it was potentially unsafe in periods of time, which I know a lot of units are feeling the same and I know we're essentially trying to almost work in war-like conditions where you're trying to save lives, but it is quite hard when you're used to giving really good care to a very high standard with all that holistic care.* [Theresa, May '20]

They also described the impact of patients being alone in the hospital and relatives unable to visit, increasing loneliness, stress, and communication challenges. Some participants described efforts to compensate for the restrictions by trying to enable communication with families; for example, staff used their own phones to contact families, information centres and equipment were set up to facilitate communication, and then single visitors in PPE (required to self-isolate afterwards) were allowed for dying patients.

> *. . .staff were using their own phones to phone the relative of someone who'd just been brought in with an ambulance, to say goodbye to them because they were going to die. . . .we now have*

*a supply of tablets. . . [for patients to] stay in contact because we will only allow visitors, because of risk of infection, if someone is truly terminally ill.* [Nick, April '20]

**Impact on health and wellbeing.** *Perceived risk of COVID-19 and impact on others.* Participants described different influences on their perceived risk of COVID-19. Inadequate PPE and seeing other HWs admitted with, or dying from, COVID-19 seemed to increase a sense of risk, whereas considering themselves young and healthy seemed to mitigate participants' concerns. Participants assumed that they would get (or had already had) COVID-19 at some point. Some highlighted the variety of responses among fellow HWs related to perceived risk and, consequently, decisions in how they were prepared to care for patients. Some described risk as part of their role ('vocation') as HWs but that it should be mitigated by adequate protective equipment.

*. . .we do sign up to risk and that's why in a sense the healthcare worker deaths in some ways in fact well that's what we signed up for. But I guess if I was in the military, the irritation was if you didn't have the right equipment, you'd be pissed off if you all died. Whereas if you have the right equipment, that's just what you sign up to.* [Nick, April '20]

Participants described being more concerned about posing a risk to others and accidentally passing COVID-19 at work, home or a supermarket. Some discussed other HWs staying in hotels to protect their families but considered the unknown timescale of the situation as a challenge to such a decision.

*Tiredness and emotional weariness.* With the increased stress and working hours by mid-April participants described tiredness and emotional weariness among HWs. Some highlighted awareness that this might become a long-term challenge as they expected the pandemic to continue to put pressure on the healthcare system.

*It's hard and [staff] are getting tired. And it has only just started.* [Nick, March '20]

Hot weather in May caused discomfort and dehydration among HWs working in full PPE, impacting on physical and mental health. Further, while participants reported that some support was available in their organisations (e.g., online resources, 'wobble'/wellbeing rooms and counselling in hospitals), they described not using these resources due to lack of time or the impracticality of accessing them during work-time. Most referred to relying on support from colleagues (e.g., professional/team 'WhatsApp' groups), families, and friends. However, some described avoiding offloading to family/friends because they were also stressed and may not understand HWs' experience.

*I couldn't even speak to someone without bursting into tears or just the despair of the situation, thinking this is like nothing I've ever–I mean I've been nursing for 30 years, I'm not new to this business. But it was pretty horrific.* [Theresa, May '20]

*Experiencing patient/public attitudes.* Participants experienced patients as understanding of limited services and grateful to HWs. They appreciated, but found unexpected, expressions of gratitude towards and support for HWs (e.g., providing free meals to hospital HWs).

*. . .you're definitely getting more of a mention or you're getting more messages of support and things like that which is really nice. . . I've been a nurse for 10 years and I don't think I've ever. . . two best friends I don't think I've ever got any kind of like 'oh do a great job' or*

*whatever, and then now they're like oh you. . . so, it's really quite funny how that's changed.*
[Dylan, April '20]

Participants described how initially in the pandemic patients seemed to limit contact with healthcare services and wanted to leave the hospital as soon as possible. Later in this phase, GPs reported more patients contacting them and being 'tetchy' about reduced care and problems with referrals.

## Phase 2: Consolidation and preparation

**Context.**  Following the peak of the first wave of COVID-19 in the UK in April 2020, the numbers of patients in emergency departments and hospitals started to gradually decline, whereas they were increasing in general practice. Participants described being 'in transition' between an emergency and a normal state–or rather 'new normal' as some measures introduced in phase 1 remained in place. This phase involved reflecting on and consolidating the experiences and lessons from the first phase, and planning and preparing for autumn/winter season driven by concerns about the potential double pressure of COVID-19 and other winter demands on healthcare system.

*I'll be okay in another environment with a week off to try and kind of regroup and get some help, get some strategies in place, and then see where we're at.* [Theresa, end of May '20]

*It feels normal actually. The only weird bit is another feeling of being in transition so are we transitioning out of being hyper vigilant down to something more relaxed so it's not static but it's not uncomfortable.* [Bonnie, August '20]

**Impact on roles and care.**  *Gradual return to usual care and responsibilities*. In this phase the main focus was on safely resuming non-COVID care, specialist referrals and elective surgeries. With the gradual return of usual care, HWs across all settings slowly returned to their typical responsibilities, including administrative tasks and management meetings that stopped during the crisis. Although GPs reported little change in their roles and continuing with primarily remote triage and consultations, they discussed seeing more patients face-to-face if needed (after triage). Emergency departments' doctors described being back to a more usual type and amount of work, continuing with long clinical shifts. In hospitals, redeployed participants started returning to their old roles, and specialty training was later reintroduced. However, some described the changes as slow due to being 'on standby' in case COVID-19 cases surged again, and were concerned about limited communication and access to training.

*There's been I would say much more frustration from people now when they're feeling like things have been changed aren't really being reviewed in a timely manner to change back.*
[Taren, May '20]

*Sense of professional development and improvement in services and care.* Participants reflected on their new skills and confidence developed during the emergency phase. GPs seemed more comfortable combining remote and face-to-face consultations when needed, which helped alleviate their concerns from phase 1 about providing sub-optimal care. Redeployed participants described feeling more confident with new skills and insights; for example, Dylan described realising that he was not suitable for ICU work but with the new confidence and self-awareness he subsequently applied for a new role with more community work.

Participants across settings also described some improvements in services and care resulting from the emergency phase, e.g., in IT systems and use of technology, end-of-life and

palliative care, communication between teams, and patient flows. However, some were also concerned that some of the positive changes (e.g., less burdensome paperwork) were waning.

*One of the best bits about COVID was just doing it. . . and not getting stuck around all the bureaucracy, and it feels very much like all the bureaucracy has come back as well. So one lesson that the government need to take away is that organisations drown in the paperwork.* [Linda, July '20]

Participants described patients as more willing to contact healthcare services with non-COVID issues, and to stay in hospitals longer as needed. In emergency departments and hospitals, with fewer patients with COVID-19, participants reported being able to spend more time with patients and give better care.

*I genuinely think that when the numbers got less and it was quieter we were able to do a better standard of care (. . .) when you slowed down, you gave them more encompassing, like sort of pathway of care for your patient. Things like my documentation and the notes got more detailed, more thorough because I just had more time to do it and so all those things improved.* [Kyle, May '20]

*Learning and preparing for future*. Despite some improvements, participants recognised that the pre-pandemic challenges remained or worsened during the pandemic (e.g., staff shortages, delayed referrals). They highlighted the need to reflect on and learn from the experience of the first phase/wave, reporting staff debriefs and resumed quality improvement initiatives. However, some expressed concerns whether the right conclusions were drawn.

*We did debrief today. . . Some of it was interesting because it was learning from what has been done well and what has been done badly, but as always with these senior meetings, there's an awful lot of, 'This was all done really well', and then you're always slightly careful bringing up what's done badly so that you don't piss off the powers that be. . .* [Nick, June '20]

Moreover, participants were concerned about the potential impact of COVID-19 ('*small spikes and hopefully not. . . a second wave'* [Nick, June '20]) and seasonal pressures, and the need to plan and prepare for winter. Nevertheless, planning was seen as very challenging due to many uncertainties.

Redeployed participants described the need for training for those who might be redeployed again in the next wave; although some described it as insufficient or unavailable to them. Overall, participants described feeling more prepared to manage surges of COVID-19 given their experiences, and some measures and processes established in phase 1, as well as some preparations undertaken in phase 2.

*I think that they've prepared a bit better this time. . . .the people who have been redeployed we've got our learning needs and we need to be, they need to be attended to. Whereas in the first time around, I felt that I was thrown into the deep end.* [Carmen, November '20]

**Impact on health and wellbeing.** *Frustration with working conditions*. Some reported feeling disillusioned with debriefs and frustrated that, despite the appreciation for HWs during the first phase/wave, the working conditions were not improved. They discussed how recognition of HWs was needed in terms of better resources and working conditions rather than symbolic gestures.

*I thought that they really didn't care about us. And that it wasn't very fair. And like getting shampoo and a toothbrush from charity is wonderful, it's a very good touch, because they, they bring us like goodie bags. . . which is brilliant and I really appreciate it, but I just wanna see something essential to us. I mean, I can buy shampoo, yes, it's a freebie, wow, very nice, I really appreciate it, but it's not gonna sort my actual problem.* [Claire, July '20]

*Partial recuperation.* Over the summer participants described taking short breaks from work, which helped them partially recharge and re-energise. However, insufficient staff to cover absences meant that time off work was too short to fully recover. Participants in management roles described the need to negotiate different priorities, be sensitive about what they can ask of staff, and ensure that there is enough good will and motivation among HWs.

*. . .we can't give the whole practice an extra week's leave. . . if we don't do that, we're certainly gonna face the most uncertain future we have with a bunch of staff that we've lost all the goodwill from. . . .it would be a disaster if people start deciding they're gonna phone in sick anyway because of leave that they haven't had.* [Nathan, September '20]

*Concerns about risk of complacency and the future.* Later in this phase participants described how everything felt almost normal and, consequently, reported some risk of complacency about infection prevention measures among HWs and patients. The latter part of this phase was characterised by expectations of the next wave of COVID-19. Some participants described the stress of this uncertainty and anticipation, and anxiety about the future.

## Phase 3: Exhaustion and survival

**Context.**   The third phase started with the increasing numbers of COVID-19 cases in September 2020 leading to the second wave (peaking in November 2020 and January 2021). Initially the main focus was on continuing COVID-19 and non-COVID care. However, due to many COVID-19 cases, as in phase 1, it became impossible to continue with non-COVID care. In January 2021 there were capacity concerns in hospitals, and more patients with COVID-19 presented in settings originally less affected during phase 1 (e.g., obstetrics). There was a sense of disbelief with the situation, which participants described as 'a mess', 'chaos' and 'surreal'. There was also a sense of exhaustion as HWs tried to 'survive' this phase.

*I only started [redeployment to ICU] in the first week of January and by that time it was already chaos. (. . .) It's very surreal and sometimes when I finish work and I come out to the street, and I just try to think about what has happened on the previous eight hours, it feels very surreal.* [Carmen, January '21]

**Impact on roles and care.**   *Shifting responsibilities and redeployment.* During this phase participants reported having to juggle an increasing number of responsibilities, trying to avoid the negative impacts on care that had happened in phase 1. GPs reported being very busy with high demand and workload, continuing to provide usual care, and planning and then delivering COVID-19 vaccinations. GPs and emergency departments' doctors reported dealing with ever-increasing and more severe mental health issues of patients.

Those who were (or volunteered to be) redeployed in phase 1 reported not wanting to be redeployed again due to negative experiences. However, once redeployed again, their experience seemed to be slightly better than in phase 1. For example, Carmen reported being able to work partly in her usual role and partly redeployed to ICU, administering care that she felt more comfortable with. Dylan and Claire also reported being redeployed to ICUs in January

but in more-suited roles, upskilled, and on a rotational basis which meant they could get a break from it occasionally.

*I have been upskilled to help with ICU, for example, as a dietician, mainly. (. . .) it's great for us, we do less on our computer and I guess we help [ICU nurses] out in the meantime, we eye-ball patients that we're worried about, we ask them if there is anyone who is not intubated. . . So it's quite good, I think.* [Claire, January '21]

*Balancing COVID-19 and non-COVID care.* Unlike in phase 1, participants described intending to continue with providing COVID-19 and usual care.

*The big thing that I was afraid of was missing the other stuff that was happening in the community. I don't know if it's a lesson learnt so much as an improvement first wave to second wave but we're able to filter the COVID stuff out and do the normal stuff now.* [Fiona, October '20]

However, as hospitals filled up with patients with COVID-19, participants reported that non-COVID care became impossible, specialities 'ceased to exist' and their focus was again fully on COVID-19 care. Participants described many HWs, and at times whole teams, having to self-isolate due to actual or suspected COVID-19 infections (e.g., through contact tracing). In December and January, there were concerns about capacity and under-staffing.

*There was probably a need for extra help about two weeks ago but somehow we sort of managed to get by and survive but I think there were calls for like army to come in and help out because we just needed hands.* [Dylan, January '21]

The quality and humane aspects of care were again difficult to maintain. HWs had to deal with high death rates, communicating with families about patients' death remotely, and difficult decisions balancing risks and priorities. Moreover, participants who described their areas as less or little affected in the first phase/wave reported experiencing the impact of COVID-19 more in this phase. For example, Maggie described caring for mothers and babies with COVID-19 in neonatal care, and how new measures needed to be taken to balance risks and complications. In these ways, despite preparation in phase 2, the experience of providing care in phase 3 seemed even worse than in phase 1.

*There is nothing that can prepare you for what is happening now. . . Now it's totally unmanageable.* [Carmen, January '21]

*Concerns about long-term impact of unceasing pressure.* In late January 2021 participants reported declining numbers of patients and easing of the pressure on healthcare services. At the same time, they recognised the need for measures to limit and manage COVID-19 to remain in place longer-term and to consider COVID-19 as endemic rather than an emergent disease.

*Challenge? I think it's going to be neo-normalisation. . . getting used to managing endemic COVID and non-COVID services at the same time.* [Nick, end of January '21]

Participants expressed concerns about the impact of continuing long-term high pressure and workload on HWs and healthcare services, and the necessity of some easing of the

pressure and rest for HWs ('*People need a break, otherwise people will really break.*' [Nick, January '21]).

*February is a time where we normally begin to see the end of a winter surge, but in the last few years, there's been no difference between winter and summer. Last year obviously has skewed that and been different totally, but the worry for me is that we don't get that down-time and we just carry on at the pace that we're at now, because I think we won't survive next winter . . .if we don't get some down-time. . . if people don't have an opportunity to have a proper holiday, and going into next winter, I think it absolutely is going to have issues for the team.* [Linda, February '21]

**Impact on health and wellbeing.** *Changed perceptions of risk of COVID-19.* During this phase, participants again reported different factors affecting their sense of personal risk of COVID-19. They perceived the risk as lower due to improved PPE and COVID-19 testing. Most had COVID-19 at some point during the year and this, together with positive antibody test results, made them feel safer. Later in winter, receiving COVID-19 vaccines also contributed to a sense of lower risk. However, they also acknowledged that it was still unclear how to best protect HWs. A higher sense of risk was reported by a participant after their birthday (making them more aware of the age-related risk) and by two participants who became pregnant and thus started to protect themselves by working from home.

*Exhaustion and resignation.* Under-staffing, many patients, and high workload led to exhaustion during this phase. Participants' mental health was also impacted by the traumatic experience of phase 1, worry about being redeployed again, and seeing many patients die. For example, Theresa described 'flashbacks' from the first redeployment, problems sleeping, and anxiety. While in phase 1 HWs were mobilised by the 'adrenaline' of responding to the emergency, during this phase participants described feeling exhausted, and some–depressive.

*. . .where last time we were – you get the adrenaline going, you feel you've got work to do, and everyone was going above and beyond and more, but I think now, I'm sure we'll do that again, but I think the difference is we're really tired. Really exhausted. A lot of us have very bad memories of that time and don't really want to go back to that period of time, and I think that's the big worry.* [Theresa, October '20].

*I feel emotionally and physically exhausted. My patient died on Monday, he was only 44 and then it's just very tough.* [crying] (. . .) *It's very tough and it's affecting us. I think we're all really down.* [Carmen, January '21]

Participants seemed resigned, describing how they had to continue working while trying to endure—'self-preserve' and 'survive'. Unlike in phase 1, during this phase some participants talked about needing or taking up more formal support, such as counselling.

*I've reached the point that I do need [support] because I'm waking up in the middle of the night thinking of my patients. . . There is emotional support but I'm not sure why I'm not accessing it. I think I need to do it now. . . I need to start off-loading all of this because I'm feeling quite burnt out. I just need to find the time and the space to do it.* [Carmen, January '21]

*Experiencing changed patient/public attitudes and behaviour.* Finally, participants described experiencing a change in patients' attitudes and behaviour: from the initial understanding and appreciation during the first phase/wave, patients increasingly complained about delays, limited services and were less concerned ('blasé') about COVID-19 and preventive measures

during the second wave (phase 3). Participants expressed feelings of frustration with the public perceived as 'forgetting' about COVID-19, and the Government perceived as failing to prevent the high numbers of COVID-19 cases and hospitalisations.

> *[The Government] waited right up until that last weekend before Christmas and I think the damage was done then already. (. . .) Last time it was accepted because it came out of nowhere and people were just surprised by it. . . And then this time round, people are just breaking the rules so then that puts pressure on us and I guess then you build up this kind of different attitude towards how you care for people. You care for them all the same but psychologically you're just like 'what is this'. (. . .) So when you read about people going to lockdown parties, visiting people on Christmas and whichever, I think people are finding it harder to be as empathetic to them.* [Dylan, January '21]

## Discussion

In the first wave (phase 1) of the COVID-19 pandemic, HWs had to rapidly adapt to working in unfamiliar ways and develop new skills with little or no training or guidance. This led to high levels of mobilisation, stress, and concern about potential unintended consequences. Over the summer (phase 2), with fewer COVID-19 cases, HWs gradually returned to usual care and roles, and attempted to reflect on, consolidate and learn from their experience, prepare for future spikes of COVID-19, and restore energy and resilience. There was a sense that the 'worst' was over and of better preparation to continue COVID-19 and non-COVID care. Nevertheless, in the second wave (phase 3) HWs were already exhausted, tried to keep delivering routine care alongside COVID-19 care, and the large numbers of patients and sick/self-isolating staff was perceived to have overwhelmed the healthcare system, having a more significant impact on HWs than in the first wave. Our findings also highlighted the interconnectedness of different parts of the healthcare system, such as the impacts of fluctuating numbers of patients between primary and secondary care and the availability/lack of specialised services.

## Comparison with literature

Firstly, our findings demonstrated links between HWs' professional roles, resources and working conditions, and personal health and wellbeing. Similarly to studies early in the pandemic, we found that initially HWs were concerned about risks to themselves and others resulting from insufficient PPE, COVID-19 testing, evidence and guidelines [7–12]. As acknowledged by others [28–30], participants in our study recognised that HWs' vocation and duty to care warrants hard work and some risks in order to save people's health and lives, but with adequate resources and working conditions to enable them to do this safely and well. While some needs (e.g., PPE) seemed to be met over time, concerns about wider under-resourcing and understaffing remained unaddressed. The pandemic has added pressure on the already-strained healthcare services which responded at a great cost to, and significantly impacted morale and resilience of, HWs. Inadvertently, the pandemic worsened the workforce crisis in the NHS with thousands of HWs expressing intentions to leave the profession due to dissatisfaction with pay/benefits, issues related to COVID-19, work-life balance and mental health [31, 32].

Previous studies also highlighted the negative impact of working during the pandemic, with unprecedented levels of uncertainty and pressure, on HWs' mental health [11, 33, 34]. Our study showed how (among other factors) changes to professional roles and care provision contributed to this, and further how these negative effects have accumulated over time, with HWs' resilience more significantly depleted during the later stages of the pandemic. Participants

described the availability of professional/organisational support but often reported not using it (e.g., due to lack of time); they needed resources, such as more staff, to enable them to deliver services and quality of care they would like to have delivered. This echoes another study showing that HWs experienced barriers to accessing support and found structural conditions of work a more significant influence on their wellbeing than individual-level strategies [13].

Secondly, our study illustrated the interconnectedness of the healthcare system and risks of unintended consequences in healthcare, in line with calls to embrace complexity and a systems approach to healthcare [35, 36]. Participants' accounts revealed the impacts of changing patient/public behaviour and Government policies, and fluctuations in patients (numbers, presentations) and service provision across settings and over time; e.g., when GPs reported seeing fewer patients, emergency and secondary care HWs reported more patients in phase 1, and vice versa in phase 2. During phase 1 HWs were very concerned about negative consequences of the changes to usual care that prioritised reducing in-person contact, freeing up resources and COVID-19 care. After that, the main concerns shifted to attempting to return to and continue usual care, especially for chronic conditions and preventative care, alongside COVID-19 care. However, this was often difficult across all settings; in primary care because of the increasing backlog and complexity of patients [37], in secondary care due to some hospitals reaching almost full capacity with patients with COVID-19 and insufficient staff [38, 39]. While there is already emerging evidence on 'collateral damage' resulting from changes and delays to non-COVID-19 care [40–42], with time we will understand the wider and longer-term impacts of the limited non-COVID-19 care.

Thirdly, the value of learning from HWs' experiences of the pandemic has been highlighted [4–6], and our study contributes longitudinal evidence to this. There are many potential 'blind spots' (including 'general complacency' and 'ignoring simple and effective non-pharmacological measures) which can be addressed by proactive measures and learning from others [43]. Insufficient preparedness for the pandemic initially caused notable stress and pressure for HWs [9, 12], followed by calls for reflection and learning from the experience. Better communication and training for HWs taking on new responsibilities (e.g., redeployment) was one identified area for improvement, as highlighted by others [37]. Participants also recognised some positive impacts on their professional development and service delivery (e.g., IT systems, workflows). Similarly, others showed that the pandemic demonstrated that quicker changes in the healthcare system are possible [7], and that the experience also prompted personal growth, reflection, and a sense of purpose among HWs [8]. We found the need for planning and preparation for winter, reflected more widely [21–23], and hope that the situation (and number of COVID-19 cases) in future would be better than in phase 1. Yet, the subsequent surges of COVID-19 were even larger, putting even more pressure on HWs and healthcare services. The reasons for this enhanced pressure, and potential lack of learning (or its implementation) from the first wave, should be explored and addressed as part of a future pandemic strategy. The House of Commons Health and Social Care and Science and Technology Committees are conducting a joint inquiry into lessons learnt from the COVID-19 pandemic with witnesses from NHS England and Improvement referencing lessons learnt within the health system [44]. The types of, and the speed of identifying and acting upon, lessons may differ, but it is important that the on-the-ground experiences of frontline HWs are also taken into account at every level of planning and workforce management in the UK and internationally.

## Implications

The governments, policy-makers, advisers and leaders in healthcare need to identify, consolidate, and implement the learning from the first year of the pandemic, strengthen the

healthcare systems in terms of enhanced human, process and practical resources so that they can better cope with demands of the pandemic, and avoid placing intolerable burden over long periods on individual HWs. This learning should come not only from leaders and decision-makers but also from the experiences of frontline HWs. Our findings suggest the importance of and need for: (i) measures to prevent hospitals from reaching or exceeding their capacity and adequate resources to continue COVID-19 and non-COVID care and address the backlog of elective care; (ii) strategies to protect HWs' mental health, prevent burnout and/or leaving the profession, and allow HWs to recover from the pressure and stresses of the pandemic; (iii) support that is accessible to HWs (e.g., when needed, outside work) and includes resources (e.g., staff, equipment) and structural changes (e.g., better communication, redeployment processes, skill-mix, training); (iv) prompt and responsive training for HWs as their and their patients' needs and concerns change [11, 37]; (v) planning adequate staffing longer-term, and ways to retain and recruit HWs; (vi) better outbreak preparedness (e.g., ensuring PPE, HWs trained/skilled for emergency redeployment), response (e.g., lessons on communicating and implementing rapid changes in complex systems), management (e.g., effective measures to reduce transmission), and retaining improvements post-pandemic (e.g., efficient workflows, remote and in-person care, IT functionality). Our findings show that the context, experiences, and concerns of HWs change during a pandemic, and different priorities need to be addressed at different phases. Ongoing attention is needed to how these change in future in the context of COVID-19 and other health emergencies; longitudinal data should be collected as this can provide unique insights into the impacts of and responses to the pandemic. Finally, COVID-19 has already prompted a debate that needs to continue about ethical issues concerning healthcare provision, risk management, and palliative care.

## Strengths and limitations

Using a longitudinal qualitative study design enabled investigating change over time and collecting more in-depth data [24]. Conducting serial interviews with the same participants over a year, allowed us to develop good relationships and trust with participants and, thus, helped collect in-depth, honest personal accounts; it also necessitated a closure which was facilitated by 'debriefing' calls and emails at the end of the interviews [24, 45]. Some participants commented then that this had been an opportunity for them to reflect and process their experiences. While the professional, research 'boundary' was maintained (e.g., with semi-structured topic guides), the cathartic/therapeutic value of research interviews in health contexts is widely recognised [46, 47]. To ensure quality and trustworthiness, throughout the study we checked our understanding and interpretations with the participants; kept detailed records and notes; involved multiple researchers in conceptualisation, analysis and interpretation; and triangulated our findings with parallel interviews with policy-makers and advisers, and with relevant publicly-available information (e.g., from select committee meetings, media/news stories). The study is reported following the standards for reporting qualitative research (the checklist is in S2 Table)[48].

Participants were rapidly recruited at the start of the pandemic using snowball sampling and our networks for promptness and convenience so their representativeness may be limited. While we included participants from different areas and organisations, our sampling strategy and relatively small number of participants meant that we did not explore other factors that might have influenced HWs' experience (e.g., local prevalence of COVID-19, resources, lockdowns or public behaviour). We included participants in different roles and settings, but sample heterogeneity posed a challenge to identifying and reporting all commonalities and differences. Limited budget allowed us only to transcribe selected interviews. To ensure

prompt analysis of the large dataset, the analysis relied on detailed notes and data coding was approached deductively. However, these are pragmatic and efficient approaches common in rapid research during public health emergencies [4, 5].

## Conclusions

HWs' experiences and concerns changed throughout the first year of the COVID-19 pandemic: from a rapid mobilisation in response to an emergency, then a partial recovery, consolidation and preparation, finally to a brink of capacity, exhaustion and survival. Our findings show the interactions between HWs' professional roles and personal health and wellbeing, the need for adequate NHS resourcing and working conditions to enable them to provide good care and avoid burn-out, and the interconnectedness of the different healthcare services, public/patient behaviours and Government decisions. This article illustrates how we can learn from the long-term experiences of the COVID-19 pandemic, and highlights the importance of supporting the resilience of HWs and addressing their changing needs during different phases of the pandemic and other health emergencies. It also highlights the need to identify and implement the lessons learnt, and strengthen the system-level resilience with resources and processes to enable healthcare services to respond to current and future demands and emergencies.

## Declarations

### Ethics

The study was approved by the University of Oxford's Medical Sciences Interdivisional Research Ethics Committee (ref. R69302). All participants gave informed consent verbally to participate in the study; a written record of each participant's consent was made, with copies retained by the participants and researchers.

## Supporting information

**S1 Table. Interview topic guide.**
(PDF)

**S2 Table. Standards for Reporting Qualitative Research (SRQR) checklist.**
(PDF)

## Acknowledgments

We would like to thank the participants for their valuable time and input, especially during such a demanding period.

## Author Contributions

**Conceptualization:** Aleksandra J. Borek, Caitlin Pilbeam, Hayley Mableson, Marta Wanat, Paul Atkinson, Sally Sheard, Anne-Marie Martindale, Tom Solomon, Christopher C. Butler, Nina Gobat, Sarah Tonkin-Crine.

**Data curation:** Aleksandra J. Borek, Caitlin Pilbeam.

**Formal analysis:** Aleksandra J. Borek, Caitlin Pilbeam.

**Funding acquisition:** Sally Sheard, Tom Solomon, Nina Gobat, Sarah Tonkin-Crine.

**Investigation:** Caitlin Pilbeam, Hayley Mableson.

**Project administration:** Aleksandra J. Borek, Caitlin Pilbeam.

**Supervision:** Sarah Tonkin-Crine.

**Writing – original draft:** Aleksandra J. Borek.

**Writing – review & editing:** Caitlin Pilbeam, Hayley Mableson, Marta Wanat, Paul Atkinson, Sally Sheard, Anne-Marie Martindale, Tom Solomon, Christopher C. Butler, Nina Gobat, Sarah Tonkin-Crine.

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
