## [Decision Letter · Decision Letter 0]

21 Oct 2021

PONE-D-21-28233Experiences and concerns of health workers throughout the first year of the COVID-19 pandemic in the UK: a longitudinal qualitative interview studyPLOS ONE

Dear Dr. Borek,

Thank you for submitting your manuscript to PLOS ONE. After careful consideration, we feel that it has merit but does not fully meet PLOS ONE’s publication criteria as it currently stands. Therefore, we invite you to submit a revised version of the manuscript that addresses the points raised during the review process.

We look forward to receiving your revised manuscript.

Kind regards,

Virginia E. M. Zweigenthal

Academic Editor

PLOS ONE

Journal Requirements:

Additional Editor Comments (if provided):

Dear Authors,

This is a fine piece based on research that investigated the experiences of health care workers over the course of the first year of Covid. Its strength is its longitudinal nature, that it includes a range of health care workers, working in a range of setting: hospital and GP.

Please note the reviewers’ comments including identifying the geographical location of their practice, such as whether they worked in centres with huge surges.

In addition, please review some turn of phrase such as ‘a lot of’ found in lines 172 and 551.

With a few modifications it should be ready for publication.

Many thanks,

Dr Virginia Zweigenthal

Reviewers' comments:

Reviewer's Responses to Questions

**Comments to the Author**

1. Is the manuscript technically sound, and do the data support the conclusions?

Reviewer #1: Yes

Reviewer #2: Yes

2. Has the statistical analysis been performed appropriately and rigorously? 

Reviewer #1: N/A

Reviewer #2: N/A

3. Have the authors made all data underlying the findings in their manuscript fully available?

Reviewer #1: Yes

Reviewer #2: Yes

4. Is the manuscript presented in an intelligible fashion and written in standard English?

Reviewer #1: Yes

Reviewer #2: Yes

5. Review Comments to the Author

Reviewer #1: Many thanks for the opportunity to review this article. Wow really a fascinating read. I don't have a lot of negative or even constructive comments to make to be honest. Would seem to be methodologically sound from an experienced group. I am mostly surprised by the absolute similarities in phases and experiences in myself and colleagues here in Cape Town. Some really useful takeout’s. Some points:

1) the paper is long. not sure if there is a way to shorten it. I know its ok for qual work to be long - but fascinated as I am, its quite a drawn out story which I think could be told more succinctly.

2) you have conducted an amazing number of repeat interviews with some of the participants - 10 in one case! I think this needs more discussion - the value of this methodology firstly? the changes in individuals - were they all consistent? surely the relationship with interviewer changed dramatically over time after so many interviews? into more of a therapist role? can you mention that more - you just say something briefly about the positive reflective role...

3) for a non-UK reader there is quite a focus on the NHS and perhaps pointing/ blaming. Not sure this is necessary or useful - its a global audience and we aren't all that concerned with what you think the NHS/ Uk government did right or wrong I feel.

4) I get the feeling this is a fairly geographically small area of the UK involved. This is fine. But maybe place for us whether this was a particularly hard hit area? And was it better or worse resourced to start with than elsewhere in the UK? Is there reason to believe this can be generalized to the rest of the UK?

thanks again I really enjoyed reading.

Reviewer #2: Manuscript title: Experiences and concerns of health workers throughout the first year of the COVID-19 pandemic in the UK: a longitudinal qualitative interview study (PONE-D-21-28233)

This is an observational longitudinal study of 14 healthcare workers (HW) who delivered care to patients with COVID-19 during the first year of the COVID pandemic. From a total of 105 interviews conducted throughout the year, this manuscript summarises the changing experiences and challenges for HW from February 2020 to February 2021 in a qualitative manner.

The authors identified three different phases the HW experienced, first the emergency and mobilisation phase characterised by changes in responsibilities, skills, training requirements and challenges in patient care, then a phase of consolidation and preparation with return to a more normal or ‘new normal’ way of working, as well as a focus on learning and preparing for future, and lastly a phase of exhaustion and focus on survival during the winter 20/21 where healthcare services in the UK were overwhelmed with COVID cases.

This study adds to Aughterson et al’s study, which captures a snapshot of the psychosocial impact of the COVID pandemic on HW in the UK.(1) The themes they identified are, however, similar to those highlighted in this study, such as changing work conditions (e.g. 168-171) and communication challenges (line 196). Positive changes as described in Aughterson’s study were also found by the authors, but these seemed to wane over time (e.g. lines 311-314).

The longitudinal nature of this study identifies changes in the perceptions and attitudes of HW. It confirms what has anecdotally been experienced by many HW who were involved in patient care during the first year of the COVID-19 pandemic, namely that repeated challenges, such as changes to roles and work environments, as well as concerns over personal health and wellbeing over time adversely affect HW’s ability to deliver care and maintaining health and emotional wellbeing. This is particularly relevant as the UK is facing another winter of rising COVID cases and an additional high predicted incidence of influenza.

Furthermore, this study highlights how changing public attitudes and behaviours, perhaps related to misinformation through social media, but also missed opportunities relating to public health policy during the first wave and biased academic reporting, can affect the mental health and wellbeing of HW. (2)

I feel this interesting and important manuscript could benefit from the addition of regional information of the participants’ workplaces: Table 1 details their characteristics, but does not include information on where in the UK they worked. This may be of interest to the readers as experiences may have differed across the UK because of differences in access to PPE, lockdown lengths (potentially impacting on public behaviour), case load etc.

(1) Aughterson H, McKinlay AR, Fancourt D, Burton A. Psychosocial impact on frontline health and social care professionals in the UK during the COVID-19 pandemic: a qualitative interview study. BMJ Open. 2021;11(2):e047353. Published 2021 Feb 8. doi:10.1136/bmjopen-2020-047353

(2) Papadimos TJ, Soghoian SE, Nanayakkara P, et al. COVID-19 Blind Spots: A Consensus Statement on the Importance of Competent Political Leadership and the Need for Public Health Cognizance. J Glob Infect Dis. 2020;12(4):167-190. Published 2020 Nov 30. doi:10.4103/jgid.jgid_397_20

6. PLOS authors have the option to publish the peer review history of their article (what does this mean?). If published, this will include your full peer review and any attached files.

Reviewer #1: **Yes: **Peter Hodkinson

Reviewer #2: No

---

## [Author Response · Author response to Decision Letter 0]

4 Jan 2022

RESPONSE TO COMMENTS

Experiences and concerns of health workers throughout the first year of the COVID-19 pandemic in the UK: a longitudinal qualitative interview study

Editor’s Comments 

1. This is a fine piece based on research that investigated the experiences of health care workers over the course of the first year of Covid. Its strength is its longitudinal nature, that it includes a range of health care workers, working in a range of setting: hospital and GP.

Thank you for this positive comment. 

2. Please note the reviewers’ comments including identifying the geographical location of their practice, such as whether they worked in centres with huge surges.

We have addressed this comment below and in the manuscript. Please see Table 1 in manuscript at page 7. 

3. In addition, please review some turn of phrase such as ‘a lot of’ found in lines 172 and 551.

We have changed the phrase - please see changes to the manuscript at line 172 (p. 10) and line 553 (p. 26).

Reviewer 1 Comments

1. Many thanks for the opportunity to review this article. Wow really a fascinating read. I don't have a lot of negative or even constructive comments to make to be honest. Would seem to be methodologically sound from an experienced group. I am mostly surprised by the absolute similarities in phases and experiences in myself and colleagues here in Cape Town. Some really useful takeout’s. 

Many thanks for the positive comment. 

2. The paper is long. not sure if there is a way to shorten it. I know its ok for qual work to be long - but fascinated as I am, its quite a drawn out story which I think could be told more succinctly.

In this paper we describe the nuances and complexities of participants’ experiences reflected in our qualitative data collected over time. The quotes alone take 1,728 words; however, we consider these quotes important to include to enhance the trustworthiness of our research by evidencing our findings. These quotes also provide vivid accounts directly from affected health workers and are important records of their experiences. To reduce the length of this paper, we could reduce or take out some of the descriptions of findings and/or quotes; however, in doing so, we feel the paper would lose some of its richness and we would prefer to avoid this. We tried to help readers navigate and more quickly read the paper by providing a summary table (Table 2) and using multiple-level sub-headings. We hope that the current length is acceptable by the Editor. 

3. You have conducted an amazing number of repeat interviews with some of the participants - 10 in one case! I think this needs more discussion - the value of this methodology firstly? the changes in individuals - were they all consistent? surely the relationship with interviewer changed dramatically over time after so many interviews? into more of a therapist role? can you mention that more - you just say something briefly about the positive reflective role...

We agree with the Reviewer that these are important and interesting methodological issues. Given the current length of the paper, we have limited scope to reflect on these aspects further. The value of, and considerations related to, the longitudinal qualitative research and the perceived ‘therapeutic’ aspect of participating in interviews have been discussed in other publications (we have added references). In the Discussion we previously noted that multiple interviews helped develop good relationships and trust with participants, collect in-depth data, and provide an opportunity for the participants to reflect too. We have now briefly expanded our reflections and added relevant references (#45, 46, 47) at lines 599-605 on p. 28.

4. For a non-UK reader there is quite a focus on the NHS and perhaps pointing/ blaming. Not sure this is necessary or useful - its a global audience and we aren't all that concerned with what you think the NHS/ Uk government did right or wrong I feel.

The focus on the NHS reflects the context of our study, i.e. that we heard from health workers delivering care within a national health system in the UK. We referenced other international studies (references 15-20) that showed similar impact of the pandemic on health workers’ mental health and wellbeing. Our data reflect core issues of institutional trust, governance, preparedness and financing/resourcing. We signpost key actions that decision makers (including governments) can take to better prepare for future events of this kind. In this way, we also highlight the higher-level issues that can be drawn from this work beyond the UK context.

We have amended sentences at lines 519 (p. 25), 564 (p. 26), 570 and 572 (p. 27) to remove references to the NHS/UK and indicate relevance to international context. 

5. I get the feeling this is a fairly geographically small area of the UK involved. This is fine. But maybe place for us whether this was a particularly hard hit area? And was it better or worse resourced to start with than elsewhere in the UK? Is there reason to believe this can be generalized to the rest of the UK?

Participants were from different geographical regions in the UK. To make this clearer to the readers, we have added the geographical regions in Table 1. We purposively selected participants to ensure representativeness from primary and secondary care, professional roles and seniority (reported under Methods/Participants) and did not include geographic location in these criteria. In part, this was related to the need to rapidly recruit participants in the midst of a public health emergency. 

Generalisability is a concept from quantitative methodologies, and there are long-standing debates whether and how it may apply to qualitative methodologies (e.g., https://www.ncbi.nlm.nih.gov/pmc/articles/PMC1117321/ among many others). It is unlikely that all health workers had the same experiences as the participants in our study, considering the large number of health workers, settings and (potential) influences. However, this does not undermine that some had similar experience. Our aim was to explore and share those experiences, rather than identify their prevalence and moderators (for which quantitative methods are more suitable). 

We have made the following changes to the paper: Table 1 (p. 7) now includes geographic regions where participants were from. We have also now added another sentence in the Discussion/Limitations at lines 612-615 on p. 28. 

Reviewer 2 Comments

1. This is an observational longitudinal study of 14 healthcare workers (HW) who delivered care to patients with COVID-19 during the first year of the COVID pandemic. From a total of 105 interviews conducted throughout the year, this manuscript summarises the changing experiences and challenges for HW from February 2020 to February 2021 in a qualitative manner. The authors identified three different phases the HW experienced, first the emergency and mobilisation phase characterised by changes in responsibilities, skills, training requirements and challenges in patient care, then a phase of consolidation and preparation with return to a more normal or ‘new normal’ way of working, as well as a focus on learning and preparing for future, and lastly a phase of exhaustion and focus on survival during the winter 20/21 where healthcare services in the UK were overwhelmed with COVID cases.

2. This study adds to Aughterson et al’s study, which captures a snapshot of the psychosocial impact of the COVID pandemic on HW in the UK.(1) The themes they identified are, however, similar to those highlighted in this study, such as changing work conditions (e.g. 168-171) and communication challenges (line 196). Positive changes as described in Aughterson’s study were also found by the authors, but these seemed to wane over time (e.g. lines 311-314).

(1) Aughterson H, McKinlay AR, Fancourt D, Burton A. Psychosocial impact on frontline health and social care professionals in the UK during the COVID-19 pandemic: a qualitative interview study. BMJ Open. 2021;11(2):e047353. Published 2021 Feb 8. doi:10.1136/bmjopen-2020-047353

We agree that our findings are in line with those reported by Aughterson et al. (cited reference #8). We referred to this study in the Introduction and Discussion. 

3. The longitudinal nature of this study identifies changes in the perceptions and attitudes of HW. It confirms what has anecdotally been experienced by many HW who were involved in patient care during the first year of the COVID-19 pandemic, namely that repeated challenges, such as changes to roles and work environments, as well as concerns over personal health and wellbeing over time adversely affect HW’s ability to deliver care and maintaining health and emotional wellbeing. This is particularly relevant as the UK is facing another winter of rising COVID cases and an additional high predicted incidence of influenza.

We agree with the Reviewer and included the relevance of the study findings ahead of another pandemic winter (p. 26). 

4. Furthermore, this study highlights how changing public attitudes and behaviours, perhaps related to misinformation through social media, but also missed opportunities relating to public health policy during the first wave and biased academic reporting, can affect the mental health and wellbeing of HW. (2)

(2) Papadimos TJ, Soghoian SE, Nanayakkara P, et al. COVID-19 Blind Spots: A Consensus Statement on the Importance of Competent Political Leadership and the Need for Public Health Cognizance. J Glob Infect Dis. 2020;12(4):167-190. Published 2020 Nov 30. doi:10.4103/jgid.jgid_397_20

Thank you for this helpful reference. We have added a sentence and the reference (#43) in the Discussion at lines 550-552 on p. 26.

5. I feel this interesting and important manuscript could benefit from the addition of regional information of the participants’ workplaces: Table 1 details their characteristics, but does not include information on where in the UK they worked. This may be of interest to the readers as experiences may have differed across the UK because of differences in access to PPE, lockdown lengths (potentially impacting on public behaviour), case load etc.

Thank you for your positive assessment of our paper. We have added geographical information in Table 1. We have also included a sentence acknowledging the potential, unexplored impact of other factors (see also our response to Reviewer’s 1 comment #5) in the Discussion/Limitations at lines 612-615 on p. 28.

---

## [Decision Letter · Decision Letter 1]

22 Feb 2022

Experiences and concerns of health workers throughout the first year of the COVID-19 pandemic in the UK: a longitudinal qualitative interview study

PONE-D-21-28233R1

Dear Dr. Borek,

We’re pleased to inform you that your manuscript has been judged scientifically suitable for publication and will be formally accepted for publication once it meets all outstanding technical requirements.

Kind regards,

Virginia E. M. Zweigenthal

Academic Editor

PLOS ONE

Additional Editor Comments (optional):

Dear Authors,

many thanks for your resubmission.

The reviewers and I are happy with your revisions, and look forward to the article's publication.

Many thanks,

Dr Virginia Zweigenthal

Reviewers' comments:

Reviewer's Responses to Questions

**Comments to the Author**

1. If the authors have adequately addressed your comments raised in a previous round of review and you feel that this manuscript is now acceptable for publication, you may indicate that here to bypass the “Comments to the Author” section, enter your conflict of interest statement in the “Confidential to Editor” section, and submit your "Accept" recommendation.

Reviewer #1: All comments have been addressed

Reviewer #2: All comments have been addressed

2. Is the manuscript technically sound, and do the data support the conclusions?

Reviewer #1: Yes

Reviewer #2: Yes

3. Has the statistical analysis been performed appropriately and rigorously? 

Reviewer #1: N/A

Reviewer #2: N/A

4. Have the authors made all data underlying the findings in their manuscript fully available?

Reviewer #1: Yes

Reviewer #2: Yes

5. Is the manuscript presented in an intelligible fashion and written in standard English?

Reviewer #1: Yes

Reviewer #2: Yes

6. Review Comments to the Author

Reviewer #1: Thankyou for thorough responses and improved manuscript.

Happy!

I still think its long but I agree largely with your response on that and will leave it with the editor.

To my mind could argue that some quotes be supoplementaty material still. btu no strong feeling.

Reviewer #2: The authors have fully and sufficiently addressed the comments from the previous review. Geographical information has been added.

7. PLOS authors have the option to publish the peer review history of their article (what does this mean?). If published, this will include your full peer review and any attached files.

Reviewer #1: **Yes: **PW Hodkinson

Reviewer #2: No

---

## [Editor Report · Acceptance letter]

8 Mar 2022

PONE-D-21-28233R1 

Experiences and concerns of health workers throughout the first year of the COVID-19 pandemic in the UK: a longitudinal qualitative interview study 

Dear Dr. Borek:

I'm pleased to inform you that your manuscript has been deemed suitable for publication in PLOS ONE. Congratulations! Your manuscript is now with our production department. 

Kind regards, 

on behalf of

Dr. Virginia E. M. Zweigenthal 

Academic Editor

PLOS ONE